# Sensing and Mapping the Effects of Cow Trampling on the Soil Compaction of the Montado Mediterranean Ecosystem

**DOI:** 10.3390/s23020888

**Published:** 2023-01-12

**Authors:** João Serrano, João Marques, Shakib Shahidian, Emanuel Carreira, José Marques da Silva, Luís Paixão, Luís Lorenzo Paniagua, Francisco Moral, Isabel Ferraz de Oliveira, Elvira Sales-Baptista

**Affiliations:** 1MED—Mediterranean Institute for Agriculture, Environment and Development and CHANGE—Global Change and Sustainability Institute, Universidade de Évora, Pólo da Mitra, Ap. 94, 7006-554 Évora, Portugal; 2AgroInsider Lda., 7005-841 Évora, Portugal; 3Escuela de Ingenierías Agrarias, Universidad de Extremadura, Avenida Adolfo Suárez, S/N, 06007 Badajoz, Spain; 4Departamento de Expresión Gráfica, Escuela de Ingenierías Industriales, Universidad de Extremadura, Avenida de Elvas, S/N, 06006 Badajoz, Spain

**Keywords:** livestock trampling, precision grazing, sensors, soil compaction

## Abstract

The economic and environmental sustainability of extensive livestock production systems requires the optimisation of soil management, pasture production and animal grazing. Soil compaction is generally viewed as an indicator of soil degradation processes and a determinant factor in crop productivity. In the Montado silvopastoral ecosystem, characteristic of the Iberian Peninsula, animal trampling is mentioned as a variable to consider in soil compaction. This study aims: (i) to assess the spatial variation in the compaction profile of the 0–0.30 m deep soil layer over several years; (ii) to evaluate the effect of animal trampling on soil compaction; and (iii) to demonstrate the utility of combining various technological tools for sensing and mapping indicators of soil characteristics (Cone Index, CI; and apparent electrical conductivity, EC_a_), of pastures’ vegetative vigour (Normalised Difference Vegetation Index, NDVI) and of cows’ grazing zones (Global Positioning Systems, GPS collars). The significant correlation between CI, soil moisture content (SMC) and EC_a_ and between EC_a_ and soil clay content shows the potential of using these expedient tools provided by the development of Precision Agriculture. The compaction resulting from animal trampling was significant outside the tree canopy (OTC) in the four evaluated dates and in the three soil layers considered (0–0.10 m; 0.10–0.20 m; 0.20–0.30 m). However, under the tree canopy (UTC), the effect of animal trampling was significant only in the 0–0.10 m soil layer and in three of the four dates, with a tendency for a greater CI at greater depths (0.10–0.30 m), in zones with a lower animal presence. These results suggest that this could be a dynamic process, with recovery cycles in the face of grazing management, seasonal fluctuations in soil moisture or spatial variation in specific soil characteristics (namely clay contents). The NDVI shows potential for monitoring the effect of livestock trampling during the peak spring production phase, with greater vigour in areas with less animal trampling. These results provide good perspectives for future studies that allow the calibration and validation of these tools to support the decision-making process of the agricultural manager.

## 1. Introduction

The silvopastoral ecosystem characteristic of the Iberian Peninsula, known as the Montado in Portugal and the Dehesa in Spain, is a mixed system that integrates, in the same place, agronomic crops (e.g., cereals, forage, or pastures), a tree stratum (e.g., Oak trees) and livestock [1] and has been proposed to optimise economic and environmental benefits, including income from animal production, the condition of farm woodlands, and carbon sequestration [2]. Such ecosystems play a crucial role in sustaining local communities and their economies in regions with marginal soils, by providing them with an additional income through livestock farming [3]. Traditional practices carried out in these areas are often regarded as environmentally friendly and landscape-preserving, and the fields are also considered to be of high natural value [3,4]. This ecosystem covers an estimated area of 850,000 square kilometres in the Mediterranean basin, mostly occupying areas characterised by unfavourable soil and climatic conditions [3]. Sustaining an important economic activity for local populations, while maintaining the pastures’ productivity and avoiding land degradation is a challenge that will determine the socio-economic viability and environmental conservation of these semi-arid areas in the face of climate change [3].

The economic and environmental sustainability of extensive livestock production systems requires the optimisation of soil management, pasture production and animal grazing [5], which justifies the actual research interest in animal/soil interactions [1] or in tree/soil interactions [6]. Livestock production is, however, associated with some negative environmental impacts on pasture quality or on soil attributes, becoming a precursor to degradation processes [7]. About 20% of the world’s pasture areas are degraded as a consequence of overgrazing and its associated erosion and compaction [8], where the main impact mentioned is the soil compaction by animal trampling [2]. According to Jordon [9] and Drewry et al. [10], a cow exerts a greater static pressure (160–190 kPa) on soil than a sheep (approximately 80 kPa), because of their low ratio of body weight to soil contact area [8]. In the specific case of the bovine breeds involved in this study (Mertolenga and Alentejana; with the mean weight of adult cows close to 400 kg and 600 kg, respectively), considering an approximate ground contact area of about 0.01 m^2^ per hoof [8,11], the static pressure exerted by the animals at each point of contact in grazing is approximately 100 to 150 kPa. These dynamic stresses can be significantly enhanced during the movement of the cow, when not all hooves are in contact with the soil surface [8,12]. Grazing animals can exert downward pressures on the soil surface similar or greater than those of heavy mechanical equipment [1,8,12,13]; therefore, this concern is understandable [1]. Soil compaction is considered, in general, a determinant factor of crop productivity [14], known and accepted as the factor that most negatively alters soil structure [7,15]. The hoof impact of livestock tends to cause the collapse of the larger soil pores, thus forming more small pores, increasing soil bulk density and soil penetration resistance, favouring soil compaction and, consequently, hindering the regrowth and renewal of the pasture and reducing productivity [7]. For these reasons, soil compaction is associated with serious soil degradation processes which culminate in a decrease in the soil aeration and water infiltration rate, and cause waterlogging, leading to runoff [1,14,15,16]. This process has a negative impact on the soil’s productive potential [17]. Soil type, soil moisture content and grazing management (e.g., stocking rate, stocking density or timing) are some of the factors that can accentuate the compaction resulting from animal trampling [2,7]. It is known that this risk is highest when the soil moisture content tends to increase [12,13]. Consequently, the autumn–winter and spring seasons, when practically all precipitation is concentrated, are the periods of greatest soil compaction vulnerability.

The degree of compaction of the sub superficial layers of the soil can be measured through soil penetration resistance, or the resistance of the soil against mechanical penetration (Cone Index, CI, in kPa), which consists of quantifying the resistance observed against the penetration of a body of a certain shape, usually a cone [18,19]. The field penetrometer is a rapid and easy-to-use tool when compared to the more conventional soil determinations, such as soil bulk density [19]. The CI, utilised to quantify the mechanical impedance of the soil, is considered one of the key indicators for the diagnosis of the most restrictive soil layers for root growth at depth [19]. Such evaluations offer essential information about the ease or difficulty of the growth of crop root systems [18]. According to Donkor et al. [13], Krajco [16] and Debiasi et al. [20], if the CI is higher than 2 MPa, this may lead to restrictions on root penetration and growth. However, measurements of the CI values are highly influenced by diverse soil factors: intrinsic (e.g., soil moisture, bulk density, texture and structure) and extrinsic (e.g., management system) [19]. Moreover, the results of field penetrometers depend on user operating speed (penetration rate), which is often challenging to standardise; a change in operating speed alters the force the users apply to insert the equipment rod, which, in turn, changes the result [15,18]. On the other hand, the characterisation of this and other soil properties is a difficult process due to the high soil spatial variability [14] and the interaction and combination of the factors involved, particularly in silvopastoral systems [1]. The soil’s superficial micro-variability is mainly controlled by soil and crop management practices, plant roots, wet/dry cycles and, in non-tillage systems, by surface-sealing processes [19].

In this context of high soil spatial variability, it is essential for management to take advantage of the technological developments associated with Precision Agriculture (PA) [21,22]). The objectives of PA are to optimise production by increasing yield or reducing costs, minimise the use of natural resources, reduce the environmental impact and improve soil quality [14]. To achieve these goals, many new technologies have been developed and used for sensing and mapping crops and soils [14]. Thus, the use of new technologies for the correct management of farms is important to prevent the degradation of new areas, since the use of pastures is a practical alternative for feeding ruminants and, concomitantly, for producing meat and/or milk [7]. One of the most widely used parameters for the management of soil spatial variability is the mapping of apparent soil electrical conductivity (EC_a_), which allows the farmer to identify differentiated management zones (e.g., for differential fertiliser application, soil amendment or irrigation) [22,23]. The EC_a_ is defined as the soil’s capacity to conduct electric currents, which is influenced by many physical-chemical features of the soil, including the clay content [14,24,25]. The temporal stability of this parameter has allowed the recognition of the geospatial measurements of EC_a_ as a valuable mapping tool that indicates the soil’s potential productivity [14,26]). Simultaneously, the measurement of the EC_a_ can be considered a relatively inexpensive, easy and fast technique with the potential to contribute to the identification and prediction of spatial variability of soil compaction [14,16]. However, in the recent literature, there is a lack of scientific papers regarding the relationship between these two parameters [14].

Other technologies that can be very useful for monitoring animal trampling are Global Positioning Systems (GPS collars). These collars have several applications in PA; for example, they can be used to monitor preferred grazing areas [27]. The geolocation of animals by remote sensing (RS) from satellites, at regular time intervals, allows the identification of grazing patterns and areas of higher grazing intensity throughout the vegetative cycle of the pasture [27]. These preferential grazing areas are potentially at greater risk of compaction by animal trampling, especially in periods of higher precipitation, such as winter or spring in regions with a Mediterranean climate. The use of RS imagery based on the Sentinel-2 satellite to obtain vegetation indices, namely the Normalised Difference Vegetation Index (NDVI), also proved a promising tool to express the response pattern of pastures’ vegetative vigour [7,28,29,30]. Therefore, with the vegetation indices that can be obtained from digital images and are sensitive to changes in the vegetation cover of pastures, before and after grazing, it is possible to monitor and to identify degraded areas with overgrazing, as well as areas that are arid or without vegetation, for example, in large or small fields and at different time scales [7].

This study aims: (i) to assess the spatial variation in the compaction profile of the 0–0.30 m soil layer over several years; (ii) to evaluate the effect of animal trampling on soil compaction; and (iii) to demonstrate the interest of combining various technological tools for sensing and mapping indicators of soil characteristics (CI and EC_a_), of pastures’ vegetative vigour (NDVI) and of cows’ grazing zones (GPS collars).

## 2. Materials and Methods

### 2.1. Site Description, Field Management and Sampling Scheme

The field of the study (Figure 1) is located at the Mitra experimental farm of the University of Évora in the southern region of Portugal (coordinates 38°32′10″N; 7°59′80″W). This field of *Quercus ilex* ssp. *rotundifolia* Lam. and bio-diverse pastures has been used for extensive and rotational grazing of 60 adult cows of two native breeds (“Alentejana”—25 animals; and “Mertolenga”—35 animals) [5]. Of the total grazing area (about 100 ha), 20 ha were monitored (11 ha in “Field A” and 9 ha in “Field B”). Grazing was conducted to have a mean stocking rate of about of 0.6 head.ha^−1^ in both fields (A and B) between October and December. In January and February and between April and June, no animals graze in “Field B”, while in March, no animals graze in “Field A”. More details of this grazing management system can be consulted in Serrano et al. [5].

The dominant soil type of this field is acidic and a not very fertile Cambisol [31], mainly used for mixed agrosilvopastoral systems.

The chronology of the measurements carried out in the experimental field is presented in Figure 2. In April 2018, the EC_a_ and altimetric surveys, as well as soil sampling and CI measurements, were carried out. The CI measurements were carried out again in March 2019, September, November and December 2021 and March 2022. Animal monitoring was carried out between January and May 2021. Vegetation Index (NDVI) time series reconstruction was performed throughout the 2021/2022 pasture vegetative cycle (between September 2021 and June 2022).

The monitoring area (“Field A” + “Field B”) was sampled in two phases: the first in 2018–2019 and the second in 2021–2022. In the first phase, 24 Sentinel-2 pixels “10 m × 10 m” were georeferenced for sampling in areas without trees (outside the tree canopy, OTC; Figure 3a), with 12 in each field (A and B). In the second phase, half of these areas was sampled (12 sampling pixels, 6 in each field, A and B), as well as the area under the tree canopy (UTC) closest to each of these pixels (12 sampling trees; Figure 3b).

### 2.2. Characterisation of the Climate

The climate of this Mediterranean region is classified as Csa (Köppen–Geiger classification) [32]. It is characterised by high inter-annual irregularity and low rainfall (<600 mm) that is more frequent in the autumn–winter period and practically nil in the summer [33].

The thermopluviometric diagram of the Évora meteorological station between July 2015 and June 2022 is presented in Figure 4. This also shows the monthly rainfall between July 2020 and June 2021 and between July 2021 and June 2022. The great irregularity of the rainfall distribution is evident: for example, 2020/2021 shows high accumulated rainfall in February (142 mm), October (141 mm), November (108 mm) and April (106 mm) and very low rainfall in March (18 mm), while 2021/2022 shows high accumulated rainfall in March (135 mm), October (113 mm) and December (97 mm) and very low rainfall in January (5 mm), February (10 mm) and November (15 mm). This irregularity and, especially, the occurrence of events with a high concentration of rainfall, associated with poorly drained soils, can lead to situations of flooding and, consequently, potentiate soil compaction by animal trampling.

### 2.3. Soil Apparent Electrical Conductivity (EC_a_) and Altimetric Surveys

With the aim of measuring EC_a_, a contact-type sensor (Veris Technologies, Salina, KS, USA) was utilised. Measurements at a depth of 0–0.30 m were performed in April 2018. An all-terrain vehicle was used to pull the sensor. The average speed of the vehicle was 2.0 m s^−1^; consecutive passages, spaced 10 m, were made across the field. The spatial resolution of the EC_a_ measurements was a 2 by 10 m grid, since a measurement was taken every second. A global navigation satellite system (GNSS) antenna was installed near the sensor. The obtained data were used to produce the EC_a_ map with the ArcMap module of ArcGIS 9.3 software (v10.5, ESRI, Inc., Redlands, CA, USA), after conducting a geostatistical analysis with the extension Geostatistical Analyst.

The data of the GNSS antenna were used to create the digital surface elevation model (elevation map) using the linear interpolation TIN tool from ArcGIS 9.3 and converted to a grid surface with a 1 m grid resolution.

### 2.4. Soil Sampling and Laboratory Reference Analysis

In the 24 sampling areas (Sentinel-2 pixels; Figure 3a), after measuring EC_a_, composite soil samples (comprised of five subsamples) were collected at a depth of 0–0.30 m. These soil samples were analysed for moisture content (SMC), particle size distribution (texture: sand, silt and clay content), pH, organic matter (OM) and cationic exchange capacity (CEC). The standard processes used in the laboratory were described in detail by Serrano et al. [5,6]. All maps of the soil parameters were produced after conducting a geostatistical analysis with ArcGIS using a 1 m grid resolution. Was used the inverse distance weighting (IDW) interpolation of the georeferenced data.

### 2.5. Cone Index Measurements

An electronic cone penetrometer “FieldScout SC 900” (Spectrum Technologies, Aurora, IL, USA) was used to measure the soil resistance to penetration (Cone Index, CI, in kPa) [15]. The main rules for the determination of CI values are standardised by the American Society of Agricultural and Biological Engineers (ASABE; EP542 and S313.3) [15].

In each sampling point, five CI measurements were carried out between 0-0.45 m (maximum depth allowed by the device), one in the central point of the sampling area, and one in each of its four quadrants. As suggested in other works [15], to minimise possible errors resulting from the uncertainty of maintaining a constant penetration rate during the determination, measurements were always carried out by the same experienced operator. When the insertion speed changes, the equipment registers an error, and the measurement is repeated. CI measurements were carried out in the 24 pixels in April 2018 and March 2019 (Figure 3a), and in the 12 OTC pixels and the 12 UTC areas, in September, November and December 2021 and March 2022 (Figure 3b).

After the field measurements, data processing was carried out. A preliminary analysis was conducted to remove outliers from the data set. This procedure is fundamental, since the CI is measured using portable penetrometers, with manual operation, and the roughness of the soil surface and the variation in the speed of the rod going into the soil profile can influence the results [19]. The inconsistent and unreliable readings that may occur near the soil surface due the unevenness of the soil surface, led us not to consider the readings obtained in each point at 0 m depth, an aspect also suggested by Mayerfeld et al. [2]. The mean CI value of the set of five measurements was calculated for each sampling area and each depth of determination.

Taking into account, on the one hand, the recommendations of Mayerfeld et al. [2] that soil compaction investigations in silvopastures should extend to at least a depth of 0.30 m and, on the other hand, of Pentos et al. [14] that soil compaction measured in deeper soil layers is of no practical relevance because of limitations in rooting depth, in this study, the mean CI of 0–0.10 m, 0.10–0.20 m, 0.20–0.30 m and 0–0.30 m was calculated. In shallow soils, as is the case for soils typical of the Montado ecosystem [33,34], measurements below 0.30–0.35 m may be in contact with the bedrock, as reported by Mayerfeld et al. [2].

### 2.6. Animal Tracking with GPS Collars and Data Analysis

To monitor the grazing patterns, five randomly selected cows were fitted with GNSS (Global Navigation Satellite System) position loggers (“Digitanimal”, Madrid, Spain). The tracking system consisted of a GNSS unit, a lithium battery pack, a PVC enclosure resistant to water and dust and a communication module (GPS collars) [35]. A total of four loggers were programmed to collect geolocation data every thirty minutes between 1 January and 17 March 2021, and the fifth receiver was programmed to collect geolocation data every five minutes between 6 and 19 May 2021. Data were transmitted over the “Sigfox”, a global network dedicated to the internet of things featuring low power, a long range, and small data. The devices, weighing 265 g, were adjusted to the neck of the animals using a stripe with a buckle, without affecting the animals’ movements. Figure 5 shows two patterns of animal behaviour throughout the year in the Montado ecosystem: UTC at peak summer sunshine hours (a) and in preferential grazing areas in other seasons (b).

The geostatistical analyses of the GPS collars were carried out with the ArcGIS Desktop software (v10.5, ESRI, Inc., Redlands, CA, USA). The “Optimised Outlier Analysis (Spatial Statistics)” algorithm, based on incident points, obtained by the GPS collars, creates a map of statistically significant hot spots, cold spots and spatial outliers using the “Anselin Local Moran’s I statistic”. This map, with s 5 m spatial resolution, includes 5 classes (“Not significant”, “High-High cluster”, “High-Low outlier”, “Low-High outlier”, and “Low-Low cluster”) and serves to characterise the grazing density pattern. With this tool, statistically significant spatial clusters of high values (hot spots), low values (cold spots) and outliers were identified within the dataset. The characteristics of the input feature class of the data to establish settings that produce optimal clusters were evaluated, and, automatically, (i) incident data were aggregated, (ii) multiple test and spatial dependence were corrected and (iii) a proper scale of analysis was determined. When a high positive z-score for a given feature is obtained, the features of nearby areas have similar values. The “Output Feature Class” is “High-High” or, conversely, “Low-Low”, respectively, for a statistically significant cluster of high or low values. On the other hand, the “Output Feature Class” is “High-Low” or “Low-High”, respectively, when the feature has a high value and nearby areas have low values or, conversely, when the feature has a low value and nearby areas have high values.

### 2.7. Vegetation Multispectral Measurement and NDVI Time Series Reconstruction

For this study, a multi-temporal Sentinel-2 imagery data set, free of clouds and atmospherically corrected, was downloaded from the Copernicus data hub. Band 8 (B8; NIR; 842 nm) and band 4 (B4; RED; 665 nm), both with a 10 m spatial resolution, were used to calculate the satellite vegetation index (NDVI; Equation (1)) and for the reconstruction of the mean NDVI trends (NDVI time series records). Values are the mean of the set of pixel sampling areas corresponding to “high livestock trampling” and of the set of pixel sampling areas corresponding to “low livestock trampling”.
(1)$$NDVI=\frac{B8-B4}{B8+B4}$$

### 2.8. Statistical Analysis

Descriptive statistical analysis was performed for all the evaluated soil parameters.

Regression analysis with a 95% significance level (*p* < 0.05) and the analysis of variance (ANOVA) of the data were carried out using IBM SPSS Statistics package for Windows (version 28.0, IBM Corp., Armonk, NY, USA). The multiple comparisons (Tukey’s HSD test) were applied for mean separation whenever the variables presented significant differences in the ANOVA. The specific analysis of the GPS collars and the determination of the NDVI from satellite image data were described above.

## 3. Results and Discussion

### 3.1. Soil Parameters: Spatial Variability and Relationship with Soil Apparent Electrical Conductivity

The descriptive statistics of the soil parameters of the experimental field are shown in Table 1. The low pH (5.5 ± 0.2), the low clay content (10.5 ± 1.8%) and OM (1.5 ± 0.3%) and the spatial variability are the most important features. According to the classification proposed by Pias et al. [19], the soil parameters show low spatial variability (pH and sand, with a CV < 12%) or medium-to-high variability (all other evaluated soil parameters, with a CV between 12 and 62%). This spatial variability is the basis for site-specific management [36,37,38], which, in this case of extensive animal production, corresponds to variable livestock management [5]. This spatial variability is also shown in the maps of Figure 6.

Figure 7 shows the elevation (a) and EC_a_ (b) maps. These highlight the slightly undulating topography, characteristic of this region, and the very low EC_a_ values (<5 mS.m; Figure 7b), typical of coarse-textured and dryland soils [24,35].

The relationship between soil apparent electrical conductivity (EC_a_) and the soil parameters of the experimental field (Table 2) showed positive and significant correlations with moisture (SMC) and clay and soil silt content, and negative and significant correlation with soil sand content. The reverse behaviour of clay and sand in the relationship with EC_a_ is visible in Figure 8. Figure 9 shows the significant relationship between SMC and EC_a_ and between SMC and CI: on the one hand, the positive effect of the SMC in EC_a_ and, on the other, the decrease in the CI with the increase in the SMC are visible. The positive and significant correlations of EC_a_ with the SMC, clay and soil silt content, and negative and significant correlation with soil sand content has also been verified in other works [14,24,39,40,41] and shows the interest of the measurement of EC_a_ as an expedient tool for identifying homogeneous management zones [22].

In terms of possible impacts on soil compaction [14,42], the results of this study highlight the important variability in OM (1 to 2%) and clay (7 to 14%) contents. In this regard, several studies state that soil compaction is affected by small changes in soil texture [15,16,17]. Mayerfeld et al. [2] and Nawaz et al. [17] conclude, for example, that silt loam soils with low colloid contents are more susceptible than medium or fine-textured loamy and clay soils at low water contents, while sandy soils are slightly susceptible to soil compaction.

According to Pentos et al. [14], many physical, chemical and biological properties of soils that affect soil compaction, namely SMC, OM or particle size distribution, also influence the soil’s electrical parameters, and, therefore, there is a close relation between soil compaction and soil apparent electrical conductivity. The EC_a_ maps frequently show a high degree of correlation with soil compaction and, thus, can potentially provide a rapid alternative for assessing soil compaction [14,16].

### 3.2. Cone Index (CI) of Spatial Variability

The assessment of soil compaction through the CI in spring 2018 (Figure 10a) and spring 2019 (Figure 10b) showed different vertical profiles in the two subplots under study (“Field A” and “Field B”): while in 2018 there was a trend towards higher compaction in “Field B” and only in the 0–0.20 m soil layer, in 2019, there was an inverse trend, with higher compaction in “Field A” throughout the assessed soil profile (0–0.30 m).

It can also be seen that the mean CI in “Field A” showed a similar pattern in 2018 and 2019, not exceeding 2000 kPa, while in “Field B”, the CI reached values above 2500 kPa in the 5–10 cm soil layer in 2018, not exceeding 1500 kPa in 2019. It is important to link this evaluation to the livestock management: the soil compaction measurement in 2018 was carried out near the end of April, after all the animals had been in “Field B” since the beginning of April; on the other hand, the soil compaction measurement in 2019 was carried out near the end of March, after all the animals had been in “Field A” between January and March.

Figure 11 and Figure 12 show the spatial variability in the CI in the two assessments carried out (2018 and 2019, respectively) in the various soil layers (0–10 cm; 10–20 cm; 20–30 cm; and 0–30 cm). These figures reflect the CV of 20–40% shown in Table 1. The spatial pattern is variable both in terms of depth and between dates.

The descriptive statistics of the SMC and CI relative to all data obtained between September 2021 and March 2022 are presented in Table 3. The results of the ANOVA applied to the CI measurements are also presented. This analysis shows significant differences for the variables “Date of measurement” and “Depth”, and non-significant differences for the variables “Tree canopy” and “Fields”. The compaction profiles show similar patterns for OTC and UTC for all of the four evaluation dates (Figure 13). The mean separation (multiple comparison of Tukey) for the variable “Date of measurement” showed higher CI values in September 2021, followed by March 2022, with no significant differences between November and December 2021. These results show a trend towards lower CI values as the SMC increases (Figure 14), which confirms the already presented relationship between the SMC and CI (Figure 9). The mean separation for the variable “Depth” showed higher CI values at a depth of 0.10–0.20 m, relative to the other two soil layers considered (0–0.10 m and 0.20–0.30 m).

The spatial variability in the CI found in this study during the various moments of assessment (CV of 28 to 41% in 2018; of 23 to 33% in 2019; and 44 to 54% in 2021/22), associated with the vertical variability in the soil profile (depths of 0–0.10 m; 0.10–0.20 m; and 0.20–0.30 m), shows that the measurements of the CI values are highly influenced by diverse soil factors [19]. The mean CI of 1900–2000 kPa registered in this study in 2018 and 1300–1700 kPa registered in 2019 (Table 1) are values below those limits generally considered critical for plant root growth. In the evaluations performed in 2021/2022, however, the mean CI was 2200–2300 kPa in November and December 2021, reaching very high mean values in March 2022 (>2700 kPa) and, especially, in September 2021 (>3100 kPa), showing an inverse relationship with SMC. This aspect is very important, since the vegetative cycle of dryland pastures normally begins in September (after the first autumn rains), an important phase in plant root development [42]. This trend towards higher CI values as SMC decreases, or vice versa (exponential relationship), confirms the results of other studies [15,18] and is attributed to the reduction in the cohesive forces between clay particles [43]. It should be noted, however, that this relationship between the CI and SMC restricts direct comparisons of CI values among same soils with different moisture contents [15].

In this study, in addition to the significant differences found when comparing different dates of the CI measurements, resulting basically from the change in SMC, but also from changes in the animal grazing management (an aspect discussed in the next point), it is important to note the absence of significant differences in the CI between OTC and UTC regarding the topsoil layer (0–0.30 m). Several studies report higher compaction in UTC areas [1,6]. As trees grow, their aboveground weight is transferred to the soil through surface roots, which also exert compression forces on near-by soil as they increase in diameter during radial growth [1]. However, this effect can be mitigated by the accumulation of leaves and other residues UTC [1], leading to higher levels of OM in UTC areas [34], which tends to reduce soil compaction because these open new soil channels contribute nutrients that support the soil rhizosphere [1], increasing the resistance to soil deformation by increasing elasticity [17].

### 3.3. Effect of Livestock Trampling on Soil Compaction

Another aspect evaluated in this study was the effect of livestock trampling on soil compaction. Figure 15 show the grazing density map based on georeferenced information obtained by the GPS collars. In these, five classes were used: “Not significant”, “High-High cluster”, “High-Low outlier”, “Low-High outlier” and “Low-Low cluster”.

For the statistical analysis, only two classes were considered: the “High-High cluster” (a statistically significant cluster of high values), which includes the sampling points A1, A5 and A9, and the “Low-Low cluster” (a statistically significant cluster of low values), which includes the sampling points A2, B1, B2, B4, B10 and B11. The other sampling points (A3, A6 and B8) were classified as “Not Significant”. “High-High” areas correspond to a high density of animals present in the considered period, potentially with higher livestock trampling. In contrast, the “Low-Low” areas correspond to zones with a low density of animals in the same period, potentially with lower levels of livestock trampling.

Table 4 and Table 5 show the CI of each one of the sampling areas of “Field A” and “Field B”, respectively, at the four evaluation dates of 2021/2022, OTC and UTC, and at different depths. In September 2021, due to the low SMC, some sampling areas did not allow CI measurements at all depths (the electronic penetrometer showed an error due to the excessive force requested).

Table 6 shows the results of the ANOVA applied to the CI measured in “high” and “low” grazed areas, OTC and UTC, and on the four dates of 2021/2022 at different depths. This analysis shows that the compaction resulting from animal trampling was significant OTC on the four evaluated dates and in the three soil layers considered (0–0.10 m; 0.10–0.20 m; 0.20–0.30 m). However, UTC, the effect of animal trampling was significant only in the 0–0.10 m soil layer and on three of the four dates, with a tendency for a greater CI at greater depths (0.10–0.30 m), and in zones with a lower animal presence. The compaction profiles (including the average of all measurements, OTC and UTC) show, nevertheless, a systematic tendency of greater CI values in areas with “high” livestock trampling when compared to areas with “low” livestock trampling (Figure 16).

The complexity of regularly monitoring animal grazing makes the assessment of the impact of animal trampling on soil compaction difficult. In this study, the use of GPS collars on cows introduced an important technological advancement, allowing detailed and practically continuous information to be obtained on grazing pathways, preferred grazing zones and areas of higher stocking densities. Consequently, better knowledge can be obtained about the interactions between several components of the Montado ecosystem, which is essential as a support tool for making decisions [44].

Our results show a systematic trend of a greater CI in areas with “high” livestock trampling, compared to areas with “low” livestock trampling. However, the evaluation of soil compactness of silvopastoral systems (as is the Montado) is complex due to the difficulty in determining the relative importance of trees versus livestock trampling on the soil’s physical properties [1]. This work shows that the compaction resulting from animal trampling was significant in OTC areas (in all evaluated dates and depths); nevertheless, in UTC areas, the effect of animal trampling was significant only in the 0–0.10 m soil layer. The tree canopy potentially provides a cushion between hoof and soil during grazing, which can mitigate the compaction by animal trampling [1]. These results seem to indicate the important role played by tree roots in the structural support of the soil in layers below a depth of 0.10 m, reducing the potential compaction effect that animal trampling tends to produce. On the other hand, a soil with higher levels of OM (as is the case of UTC soil [34]) generally has a better structure because OM helps create large, strong soil aggregates that help resist compaction [8]. In addition, this complexity also reflects the balance of restorative and compaction processes that may occur in a differentiated manner UTC and OTC throughout the vegetative cycle of the pasture. Besides the direct effect of trampling, livestock can indirectly change soil properties by consuming vegetation that would otherwise contribute to the availability of organic matter to soil microfauna by reducing the amount and extent of fine roots that open new soil channels for the soil rhizosphere [1]. The biological restorative action of soil decompaction provided by pasture roots and by the activity of soil mesofauna is enhanced when the pasture management favours the accumulation of phytomass in the aerial part and in the root system of the plants [42,45], which is what happened in “Field B” of this experiment (several months without grazing animals and without any sampling area with “high” livestock trampling). Potentially, rest periods from grazing also enhance soil recovery by physical processes (variations in soil temperature and moisture content) [1,9].

Another relevant aspect relates to the depth at which soil compaction occurs due to livestock trampling. The compaction was significant at a depth of 0–0.10 m on three of the four dates of evaluation in both OTC and UTC areas (Table 6). At depths of 0.10–0.20 m and 0.20-030 m, the effect of livestock trampling was significant only in OTC areas. Several studies report that the highest compaction impact caused by animal trampling is confined to the topsoil layer under wet soil conditions [8], which has practical implications because deeper soil layers (below 0.15 m [17]) are generally slower to recover from compaction [1,17]. For example, Roesh et al. [8] and Debiasi [20] report that the greatest impact by livestock hooves is limited to the top 0-0.05 m soil layer, while Sharrow et al. [1], Reichert et al. [42] and Vzzotto et al. [45] extend this influence to the top 0–0.10 m soil layer, and Mayerfeld et al. [2], Medina [12] and Donkor et al. [13] indicate that this impact can reach up to a depth of 0-0.15 m or even 0.20 m [17]. This effect depends on several factors, among them the animal load [20] or grazing intensity [2]. When the animal load is very high, an increase in compaction due to trampling may occur in deeper layers [20]. The magnitude of the differences in the CI in the top 0.20 m also appears to be variable over time; therefore, the longer-term tracking of changes in the CI may be required [2].

### 3.4. Effect of Livestock Trampling on Pastures

The impact of livestock trampling on pastures was evaluated by the NDVI time series obtained from satellite imagery (Sentinel-2). Figure 17 shows the reconstruction of the mean NDVI time series retrieved between July 2021 and June 2022, without the records affected by the existence of clouds. The values are the mean of the set of pixel sampling areas of “high livestock trampling” and of the set of pixel sampling areas of “low livestock trampling”. It is possible to observe higher NDVI values in March, which reflects the differential grazing management (“Field A” versus “Field B”). In the period of peak pasture production (April and May), the behaviour of the NDVI is inverted, with higher values in areas of “low livestock trampling”, which are potentially less compacted and, therefore, with greater vegetative vigour.

The composition and production of a range of vegetation are the primary vulnerable factors to grazing pressure [46]. In this study, the impact of livestock trampling on pastures was evaluated by the NDVI time series obtained from Sentinel-2 imagery. Although this approach has some limitations, namely, the constraint resulting from poor-quality images on cloudy days, which can occur frequently during the pasture’s vegetative cycle [28,29], the NDVI can be used to monitor the pasture’s development status since it mainly reflects the chlorophyll content, an indicator of pastures’ vegetative vigour [29,30].

The results of this study show (Figure 17) an NDVI pattern that is typical of the vegetative cycle of dryland pastures in the Mediterranean region, already presented in other works [28,29,30]. This pattern is similar in “high” and “low” livestock trampling areas and reflects the effect of the evolution of air temperature and the distribution of precipitation [30]. In Figure 17, two moments are identified where the NDVI patterns indicate significantly different behaviours in the two situations of livestock trampling (“high” and “low”): (i) “differential grazing management” in “Field A” and “Field B” (stocking rates); (ii) “pasture peak production”. Higher NDVI values were found in March in “high livestock trampling” areas, which reflects the differential grazing management (“Field A” versus “Field B”): during this month, grazing was concentrated in “Field B”, where “low livestock trampling” areas predominate, while “Field A”, where “high livestock trampling” areas predominate, was left to rest (without grazing). The fact that in March the pasture of “Field A” was not grazed, accompanied by the rise in air temperature and an important accumulated rainfall (135 mm; Figure 4), allowed the recovery of vegetative vigour, which translated into a significant increase in the NDVI. In the period of pasture peak production (April and May), the behaviour of the NDVI is inverted (Figure 17), with higher values in areas of “low livestock trampling”, which are potentially less compacted and, therefore, with greater vegetative vigour. Gao et al. [47] and Jin et al. [48] also state that climatic factors such as precipitation and temperature are responsible for inter-annual fluctuations in biomass and vegetative vigour. Psyllos et al. [3] emphasise the greater yield potential of rotational grazing between fields (grazed versus not grazed). This is, however, a complex issue, as these pastures are biodiverse, comprised of grasses, legumes, forbs and other species. Different botanical species mature at different rates [49] and show different susceptibilities to animal trampling. For example, animal trampling has a positive effect on the germination of some grass and forbs species [46]. On the other hand, certain botanical species with prostrate growth (such as legumes), are more tolerant than others (grasses, for example) to livestock trampling [50], i.e., show differential susceptibility to damage by animal trampling [49], which can result in a greater abundance of trampling-tolerant plants, affecting the composition of an herbaceous plant community [50]. The assessment of the impact of animal trampling on the floristic composition of pastures is a subject that requires future long-term investigations.

### 3.5. Mitigation of the Negative Effects of Livestock Trampling

The various studies carried out and published on the effects of livestock trampling are unanimous on the greater susceptibility to soil compaction and pasture damage during periods of high SMC. Multiple management strategies to avoid or to mitigate the negative impact of livestock trampling should be a priority [42]. According to Reichert et al. [42], two preventative measures may be implemented: (i) the use of mobile fences to impede the entry of animals into more susceptible areas in the days following the occurrence of major precipitation events; (ii) control over grazing management (stocking rates) to permanently guarantee a minimum height of pasture to ensure soil surface protection.

One of the main causes of soil and pasture degradation resulting from livestock trampling is the natural tendency of animals to agglomerate in certain areas of the pasture. Thus, an understanding of the potential environmental effects of the concentration areas is necessary to adequately consider mitigating grazing management practices [51]. Some examples of grazing management practices that most influence grazing distribution and, consequently, livestock trampling, are the location of watering sites, supplement feeders (concentrate, hay or minerals), tree shade or fence and gate placement [51]. Some compaction at gates and waterers, for example, is inevitable, but can be minimised. The traditional advice of moving waterers and feeders, fences and gates to limit livestock concentration areas seems appropriate [51]. When it is possible to identify and circumscribe these critical compaction sites, then specific improvement interventions can be carried out at only the affected areas, minimising the cost and time necessary for the operation [42]. Furthermore, temporary livestock exclusion has been shown in previous studies to be an attractive tool to improve the physical properties of soil [9]. This site-specific management fits into the perspective of *Precision Agriculture* or, in this case, of *Precision Grazing*, where several technological tools can make an excellent contribution to the definition of homogeneous management zones, the basis for implementing differentiated management strategies [22,38]. The technologies used in this study (e.g., GPS collars, cone penetrometer, EC_a_ sensor and satellite imagery) can be practical tools that allow us to monitor animal grazing patterns, the spatial and temporal variability of the physical properties of soil or the condition of pastures. All these are important factors that can help farmers to assess potential compaction due to animal trampling and to adopt sustainable livestock management systems [8].

The global technological approach proposed in this study to correlate the soil compaction variables related to the effect of animal trampling is shown in Figure 18. Future research on soil compaction in the Montado ecosystem, in addition to animal trampling, may include the monitoring of the effect of tree roots. Here, too, new technological approaches can be combined. One study by Xu et al. [52] is an example of this endeavour based on the development and validation of a new tool (a high-fidelity 3D root system morphological model) for simulating the mechanical analysis of root–soil composites.

## 4. Conclusions

The economic and environmental sustainability of extensive livestock production systems requires the optimisation of soil management, pasture production and animal grazing. All these aspects are interdependent and linked to soil compaction. The compaction resulting from livestock trampling was significant in areas OTC in the three soil layers considered (0–0.10 m; 0.10–0.20 m; 0.20–0.30 m), but in areas UTC this effect was only significant in the 0–0.10 m soil layer. These results suggest that this is a dynamic process throughout the year, with recovery cycles associated with grazing management, seasonal fluctuations in soil moisture and temperature, or the spatial variation in specific soil characteristics (namely clay and organic matter contents). The application of technologies such as those used in this study (e.g., GPS collars, cone penetrometer, EC_a_ sensor and satellite imagery) can be practical tools for monitoring animal grazing patterns, the spatial and temporal variability in the physical properties of the soil and the condition of pastures. This approach shows the opportunity provided by *Precision Agriculture* technologies, such as proximal and remote sensing, to generate knowledge, support decision making and respond to the challenge of a holistic and sustainable management system of the Montado ecosystem.

## Figures and Tables

**Figure 1 sensors-23-00888-f001:**
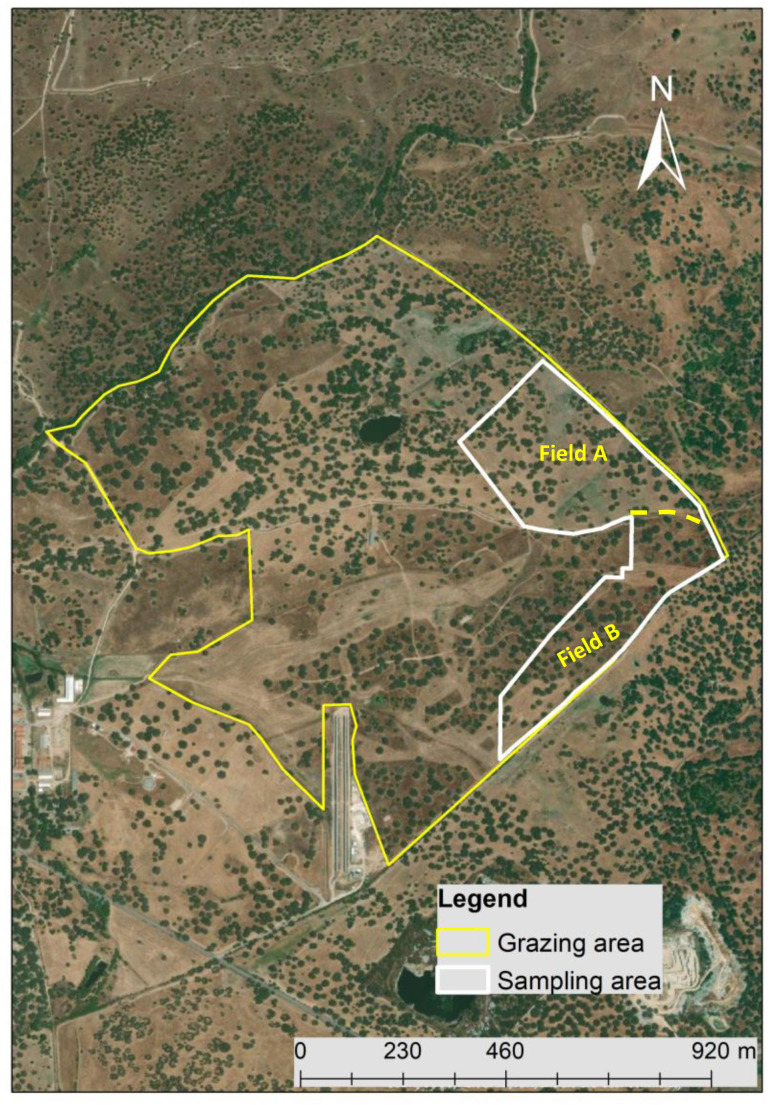
Grazing area and sampling area of the experimental field.

**Figure 2 sensors-23-00888-f002:**
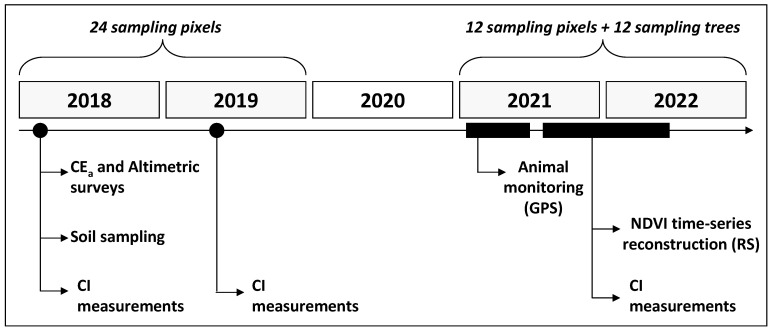
Timeline of measurements carried out in the experimental field.

**Figure 3 sensors-23-00888-f003:**
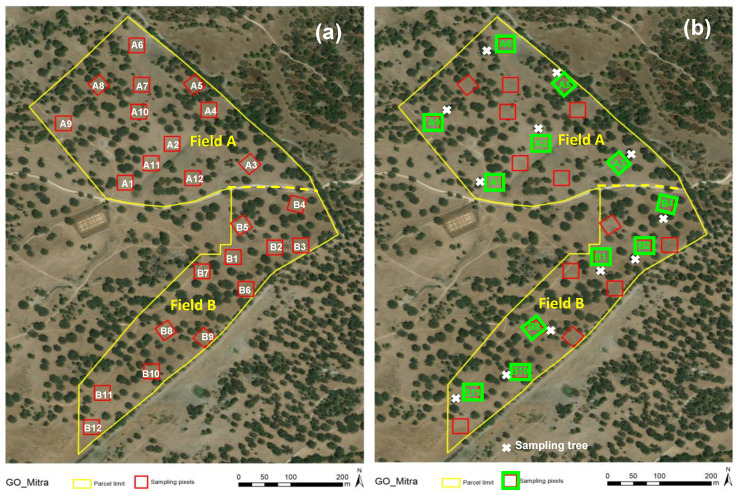
Sampling scheme of the experimental field in each phase of this study: (**a**) 24 sampling pixels; (**b**) 12 sampling pixels and 12 sampling trees.

**Figure 4 sensors-23-00888-f004:**
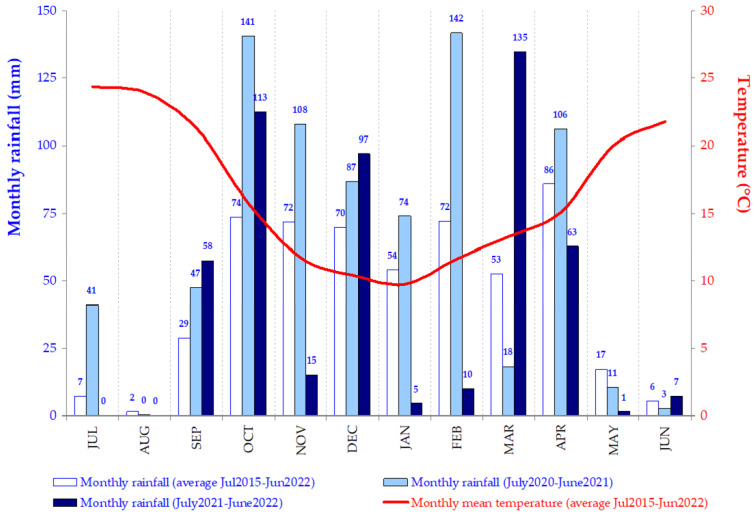
Thermo-pluviometric diagram of Mitra (Évora, Portugal) between July 2015 and June 2022.

**Figure 5 sensors-23-00888-f005:**
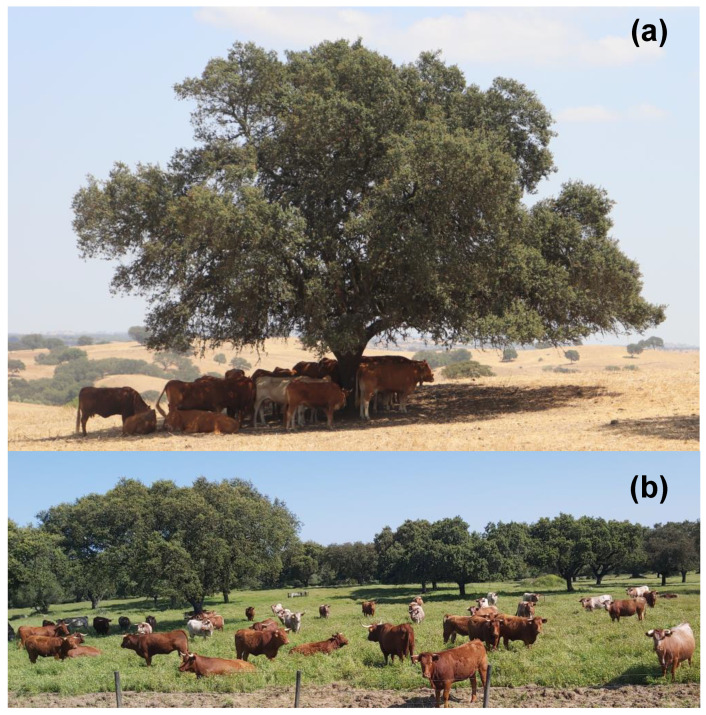
Typical behaviour of grazing cows: (**a**) under tree canopy, UTC, in the hot season (summer); (**b**) in preferential grazing areas in the other seasons (autumn, winter and spring).

**Figure 6 sensors-23-00888-f006:**
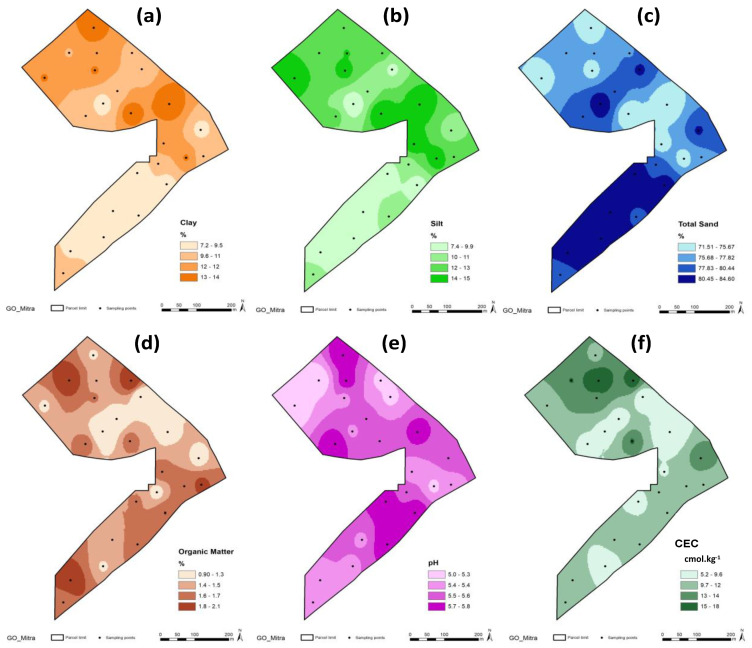
Maps of the soil characteristics: (**a**) clay; (**b**) silt; (**c**) sand; (**d**) organic matter; (**e**) pH; and (**f**) cationic exchange capacity (CEC).

**Figure 7 sensors-23-00888-f007:**
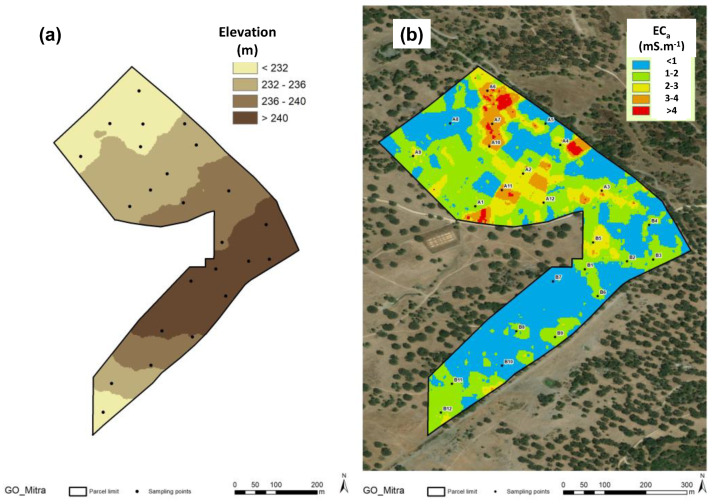
Maps of the experimental field: (**a**) elevation; (**b**) soil apparent electrical conductivity (EC_a_).

**Figure 8 sensors-23-00888-f008:**
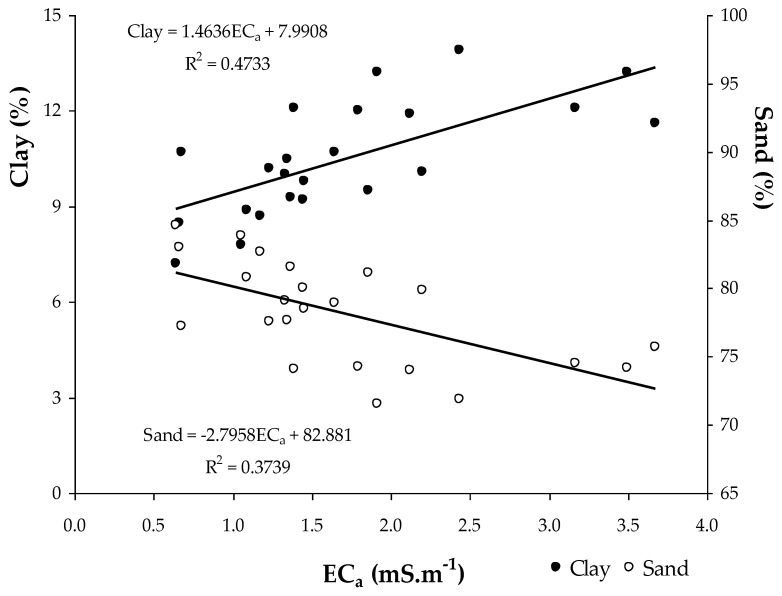
Relationship between soil apparent electrical conductivity (EC_a_) and sand and clay content in the 0–0.30 m soil layer (2018).

**Figure 9 sensors-23-00888-f009:**
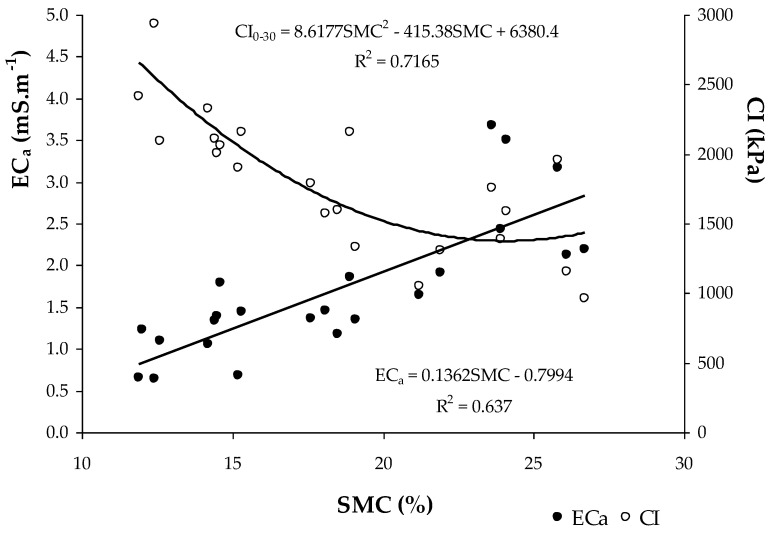
Relationship between soil moisture content (SMC) and soil apparent electrical conductivity (EC_a_) and the Cone Index (CI) in the 0–0.30 m soil layer (2018).

**Figure 10 sensors-23-00888-f010:**
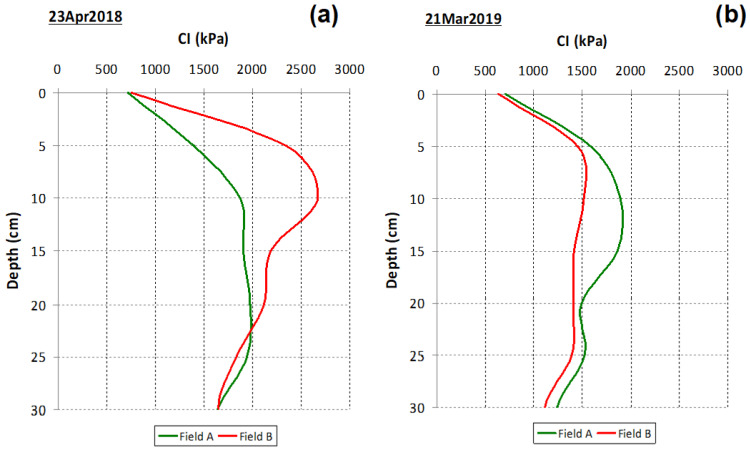
Average Cone Index (CI) at 0–0.30 m soil depth, in “Field A” and “Field B”: (**a**) 2018; (**b**) 2019.

**Figure 11 sensors-23-00888-f011:**
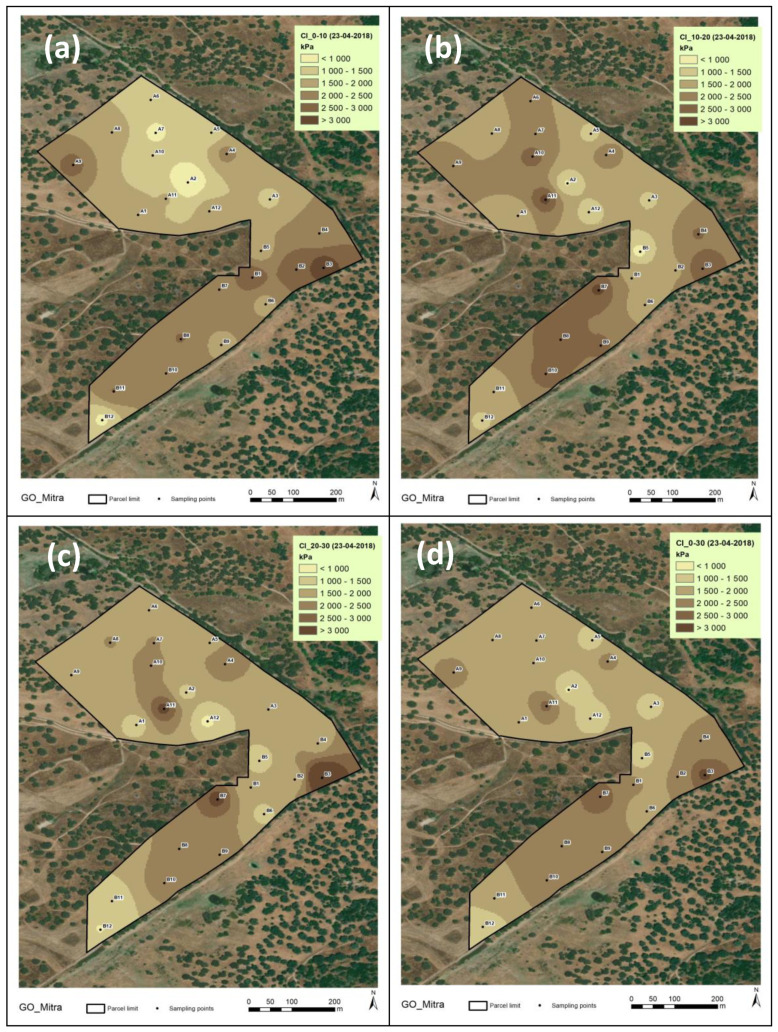
Cone Index (CI) maps of the experimental field in April 2018: (**a**) 0–0.10 m; (**b**) (0.10–0.20 m; (**c**) 0.20–0.30 m; (**d**) 0–0.30 m.

**Figure 12 sensors-23-00888-f012:**
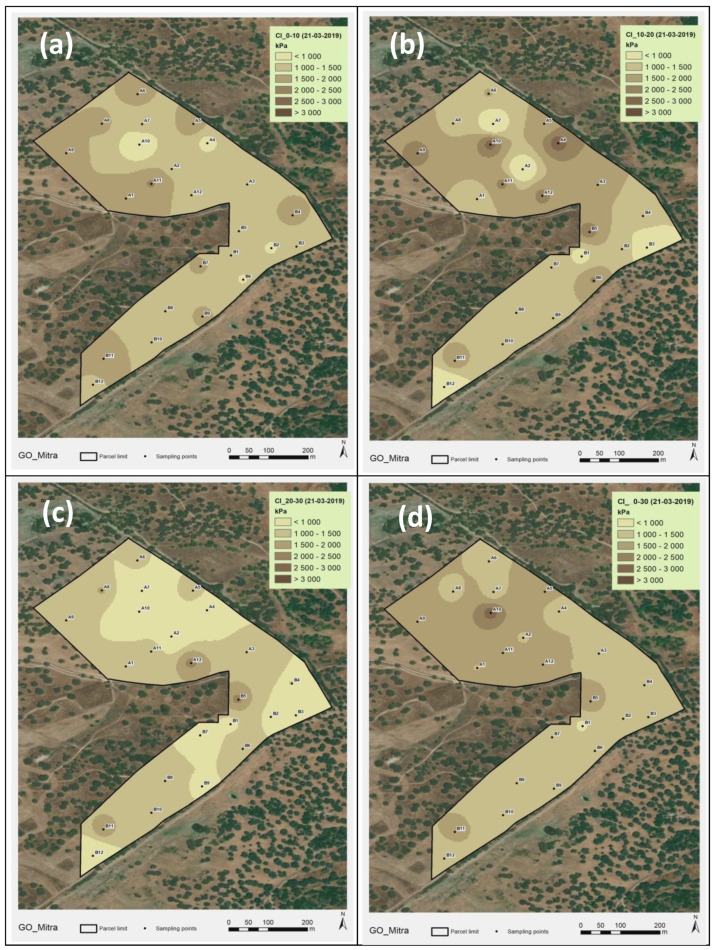
Cone Index (CI) maps of the experimental field in March 2019: (**a**) 0–0.10 m; (**b**) (0.10–0.20 m; (**c**) 0.20–0.30 m; (**d**) 0–0.30 m.

**Figure 13 sensors-23-00888-f013:**
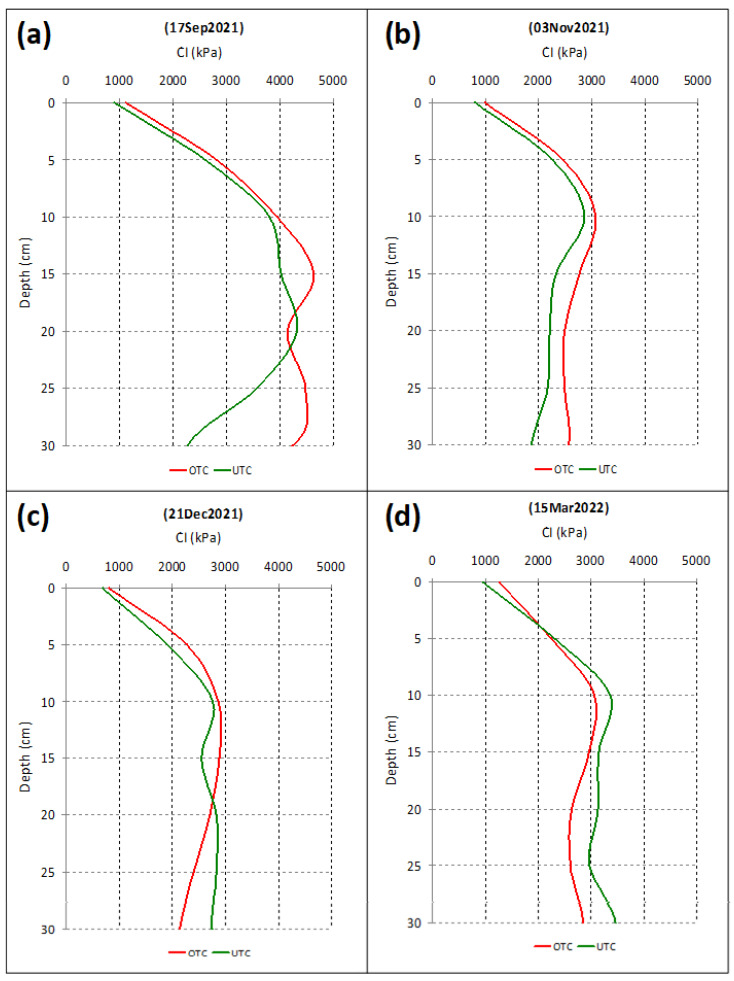
Average Cone Index (CI) at 0–0.30 m soil depth, outside and under the tree canopy (OTC and UTC, respectively): (**a**) 17 September 2021; (**b**) 3 November 2021; (**c**) 21 December 2021; (**d**) 15 March 2022.

**Figure 14 sensors-23-00888-f014:**
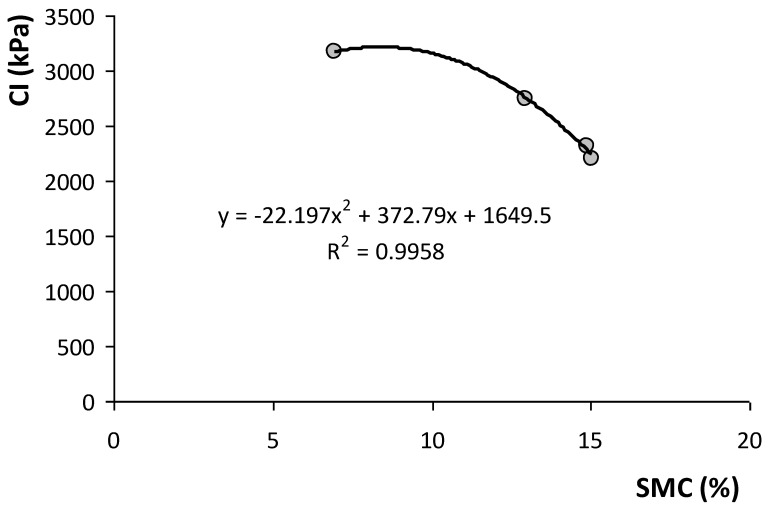
Relationship between mean soil moisture content (SMC) and mean Cone Index (CI) in the 0–0.30 m soil layer (2021/2022).

**Figure 15 sensors-23-00888-f015:**
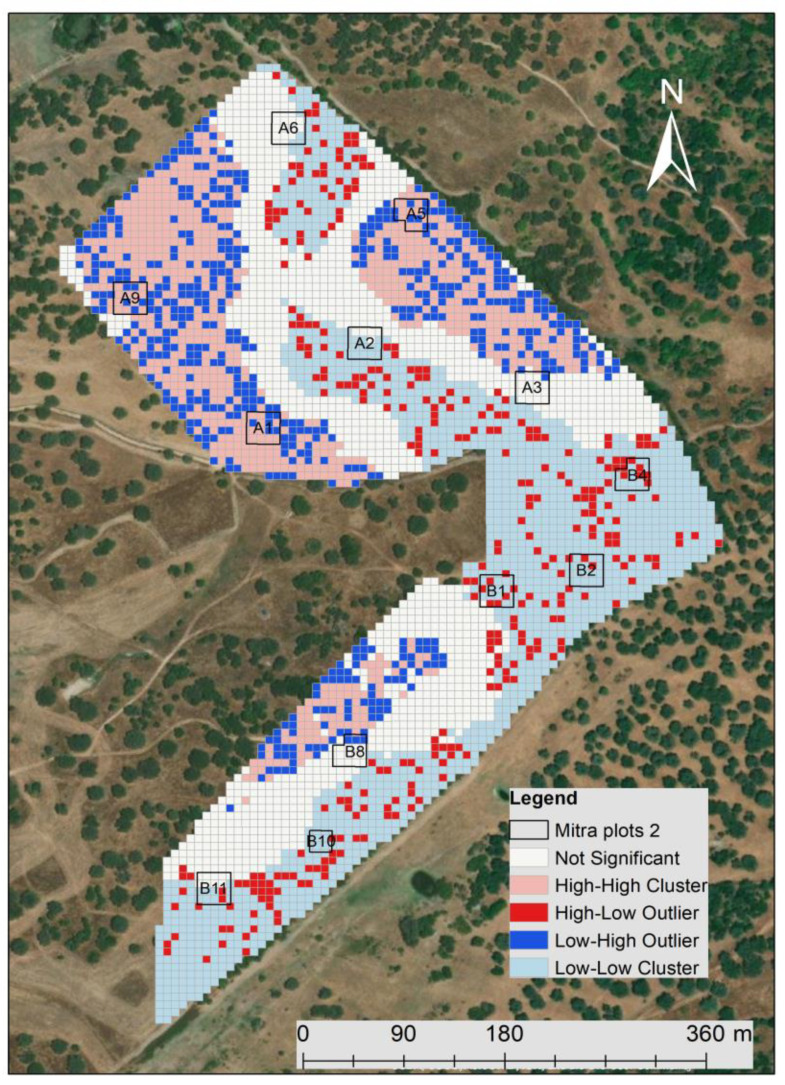
Grazing density map based on georeferenced information obtained by GPS collars between January and May 2021.

**Figure 16 sensors-23-00888-f016:**
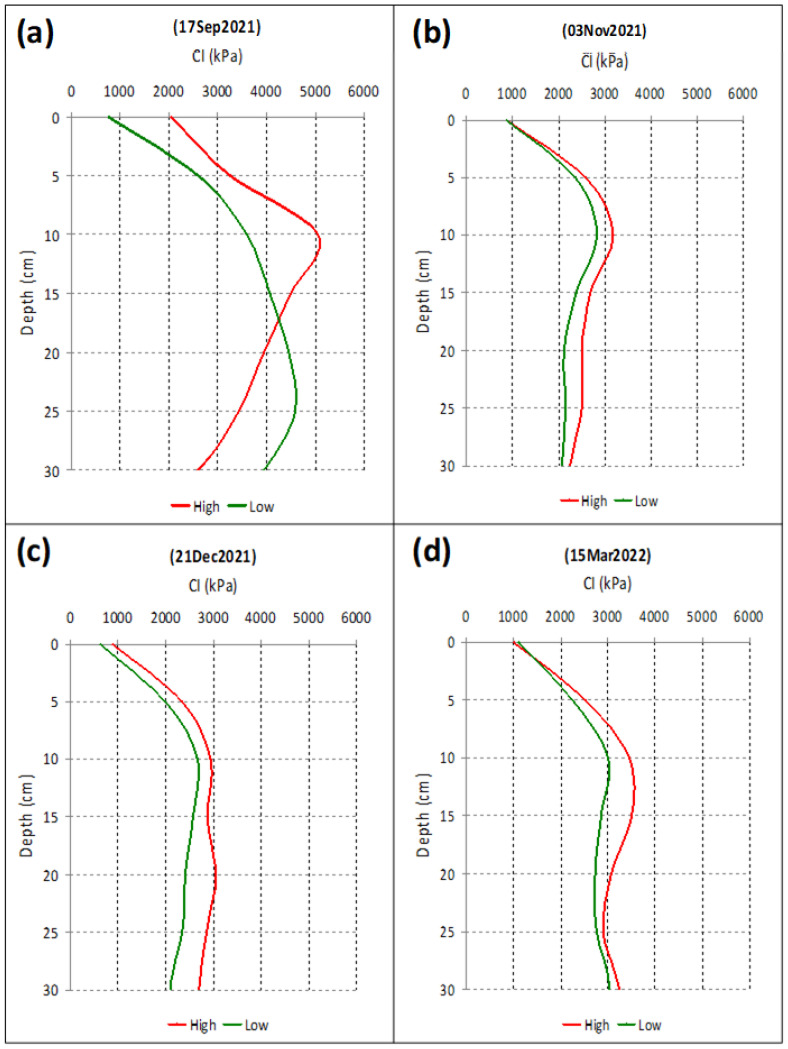
Average Cone Index (CI) at 0–0.30 m soil depth, in high and low-grazing-intensity areas: (**a**) 17 September 2021; (**b**) 3 November 2021; (**c**) 21 December 2021; (**d**) 15 March 2022.

**Figure 17 sensors-23-00888-f017:**
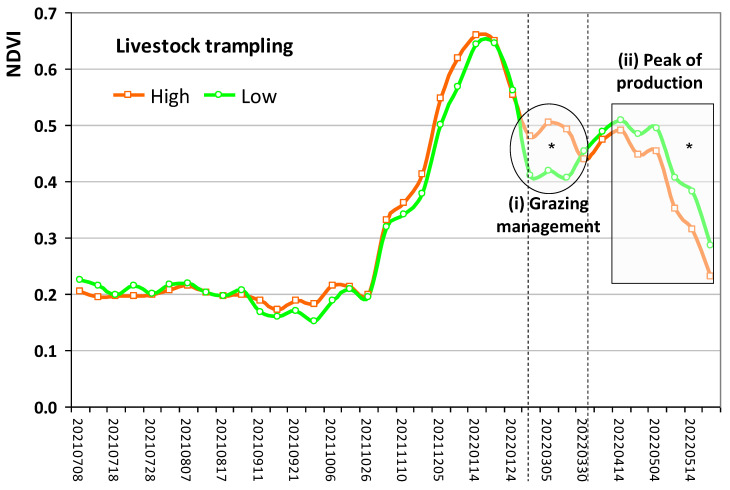
Mean Normalised Difference Vegetation Index (NDVI) time series obtained between July 2021 and June 2022 in high and low livestock trampling areas. *—Statistically significant at the 95% confidence level (*p* < 0.05).

**Figure 18 sensors-23-00888-f018:**
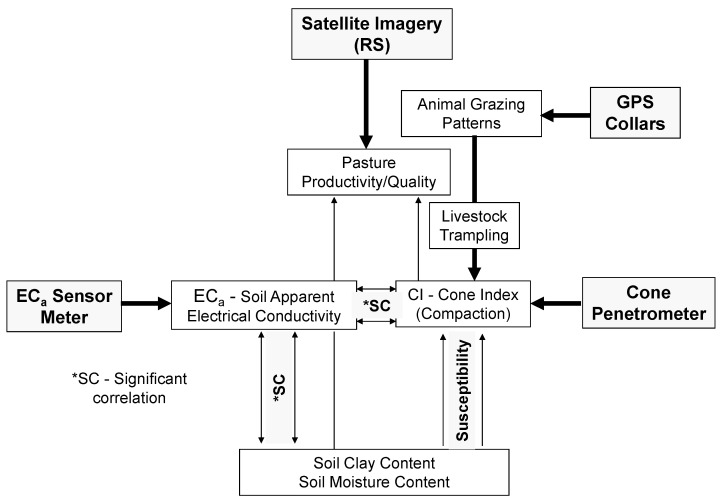
Global technological approach used in this study.

**Table 1 sensors-23-00888-t001:** Descriptive statistics of the soil parameters of the experimental field (0–0.30 m) in 2018 and 2019 surveys.

Soil Parameter	Mean	SD	CV (%)	Range
23 April 2018				
Clay (%)	10.5	1.8	17.0	7.2–13.9
Silt (%)	11.4	2.1	18.7	8.2–15.3
Sand (%)	78.1	3.8	4.9	71.5–84.6
pH	5.5	0.2	4.4	5.0–5.8
OM (%)	1.5	0.3	21.5	0.9–2.1
CEC (cmol kg^−1^)	10.8	2.8	26.4	5.2–17.9
SMC (%)	18.4	4.9	26.7	11.9–26.7
CE_a_ (mS m^−1^)	2.3	0.9	39.1	0.6–4.7
CI_0–10_ (kPa)	1875	763	40.7	207–3378
CI_10–20_ (kPa)	2042	656	32.1	853–3137
CI_20–30_ (kPa)	1912	813	42.5	896–4551
CI_0–30_ (kPa)	1970	559	28.4	207–4551
21 March 2019				
CI_0–10_ (kPa)	1441	332	23.0	845–2043
CI_10–20_ (kPa)	1687	532	31.6	744–2743
CI_20–30_ (kPa)	1313	429	32.6	331–2111
CI_0–30_ (kPa)	1445	364	25.2	331–2743

SD—Standard deviation; CV—Coefficient of variation; OM—Organic matter; CEC—Cationic exchange capacity; SMC—Soil moisture content; EC_a_—Soil Apparent Electrical Conductivity; CI—Cone Index.

**Table 2 sensors-23-00888-t002:** Correlation coefficients between soil apparent electrical conductivity (EC_a_) and soil parameters of the experimental field (0–0.30 m) on 23 April 2018.

Soil Parameter	Correlation Coefficient (Significance)
Clay (%)	0.688 (**)
Silt (%)	0.523 (*)
Sand (%)	−0.611 (**)
pH	0.227 (ns)
OM (%)	0.420 (ns)
CEC (cmol kg^−1^)	0.119 (ns)
SMC (%)	0.795 (**)

OM—Organic matter; CEC—Cationic exchange capacity; SMC—Soil moisture content; **—Statistically significant at the 99% confidence level (*p* < 0.01); *—Statistically significant at the 95% confidence level (*p* < 0.05); ns—Not significant.

**Table 3 sensors-23-00888-t003:** Descriptive statistics (mean ± standard deviation) of soil moisture content (SMC) and Cone Index (CI) between September 2021 and March 2022, and inferential analysis of CI.

Factors	n	SMC (%)	CI (kPa)	CI_Signif.
Date of measurement				*
1 (17 SEP 2021)	485	7.0 ± 1.8	3169 ± 1717	a
2 (03 NOV 2021)	1187	15.0 ± 2.6	2205 ± 974	c
3 (21 DEC 2021)	1101	14.9 ± 3.2	2318 ± 1084	c
4 (15 MAR 2022)	1106	13.0 ± 1.8	2749 ± 1197	b
Tree canopy				ns
OTC	1997	13.2 ± 3.0	2557 ± 1204	-
UTC	1882	13.6 ± 4.1	2465 ± 1252	-
Fields				ns
A	1861	14.7 ± 3.6	2506 ± 1258	-
B	2018	12.1 ± 3.1	2519 ± 1201	-
Depths				*
0–0.10 m	1384	-	2459 ± 633	b
0.10–0.20 m	1269	-	2964 ± 891	a
0.20–0.30 m	1226	-	2780 ± 966	b

n—Number of CI measurements; SMC—Soil moisture content; CI—Cone Index; OTC—Outside the tree canopy; UTC—Under the tree canopy; Signif.—Significance; *—Statistically significant at the 95% confidence level (*p* < 0.05); ns—Not significant; Different lowercase letters in the interactions indicate significant differences in the mean CI for the “Fisher’s” test (Prob. < 0.05).

**Table 4 sensors-23-00888-t004:** Mean Cone Index (CI; kPa) of the sampling areas of Mitra “Field A” on the four evaluation dates (2021/2022), outside and under the tree canopy (OTC and UTC, respectively), and at different depths.

Sampling Area	A1		A2		A3		A5		A6		A9	
Animal Trampling	High		Low		NS		High		NS		High	
	OTC	UTC	OTC	UTC	OTC	UTC	OTC	UTC	OTC	UTC	OTC	UTC
Date 1 (17 September 2021)												
0–10 cm	4079	4531	2246	3083	1636	3393	2688	3172	3243	1699	2852	3484
10–20 cm	4908	x	x	x	1000	x	4542	x	x	2656	3217	3628
20–30 cm	x	x	x	x	x	x	2661	x	x	x	4547	2171
Date 2 (3 November 2021)												
0–10 cm	2944	2354	3150	2018	2179	2259	2605	2205	1697	2950	1690	3274
10–20 cm	4090	2179	2461	1811	2622	2366	3284	2464	3660	2633	1810	2101
20–30 cm	3946	1788	1820	2061	2753	1639	2984	2474	3628	1814	1972	1493
Date 3 (21 December 2021)												
0–10 cm	2593	2580	2441	2219	2111	2705	2666	1754	1196	2449	2638	1828
10–20 cm	4065	3286	2004	2484	3272	3611	2952	2586	3033	2751	2603	1952
20–30 cm	3806	3693	1256	2165	2900	3660	2201	3003	2811	3420	1328	1966
Date 4 (15 Macch 2022)												
0–10 cm	2783	3277	2278	2544	1984	2826	3359	1779	1059	3003	3244	1588
10–20 cm	4975	4582	2366	2225	2734	3161	3175	2449	1866	4270	2751	2172
20–30 cm	4196	4295	2245	2835	2840	2980	2954	3369	2198	3816	1906	1815

NS—Not significant; OTC—Outside the tree canopy; UTC—Under the tree canopy; x—“Excessive force”.

**Table 5 sensors-23-00888-t005:** Mean Cone Index (CI; kPa) of sampling areas of Mitra “Field B” on the four evaluation dates (2021/2022), outside and under the tree canopy (OTC and UTC, respectively), and at different depths.

Sampling Area	B1		B2		B4		B8		B10		B11	
Animal Trampling	Low		Low		Low		NS		Low		Low	
	OTC	UTC	OTC	UTC	OTC	UTC	OTC	UTC	OTC	UTC	OTC	UTC
Date 1 (17 September 2021)												
0–10 cm	3175	3579	1872	1907	3300	2626	2905	1937	3582	1849	2428	2317
10–20 cm	4134	4260	x	3851	4567	5204	4646	x	3420	4640	4907	3524
20–30 cm	4975	3312	x	3420	2714	4692	4783	x	4027	1276	5899	3880
Date 2 (3 November 2021)												
0–10 cm	2579	2475	2149	2240	3030	1667	2694	2208	2351	2084	2337	1903
10–20 cm	2383	2800	1657	2196	2627	2291	3064	2182	2757	2984	1870	2192
20–30 cm	1992	2061	1587	2017	2133	1982	2932	2214	2363	3020	1972	1871
Date 3 (21 December 2021)												
0–10 cm	2512	1920	2521	1754	2236	1544	2938	1995	2518	1595	1699	2093
10–20 cm	2303	2751	2580	2314	2546	2857	3375	1627	2612	2765	2190	2751
20–30 cm	2108	2640	1883	2367	2082	2837	3227	1295	2392	3798	1955	2179
Date 4 (15 Macch 2022)												
0–10 cm	2454	2426	2809	1404	2451	2909	3187	2259	2530	2145	1515	3136
10–20 cm	2300	3898	1958	2375	2817	3776	3562	2265	3085	3361	2970	2888
20–30 cm	2441	2981	1345	3151	2515	4055	3562	3062	2581	4077	3607	3614

NS—Not significant; OTC—Outside the tree canopy; UTC—Under the tree canopy; x—“Excessive force”.

**Table 6 sensors-23-00888-t006:** Descriptive (mean ± standard deviation; kPa) and inferential analysis of Cone Index in “high” and “low” grazed areas, outside and under the tree canopy (OTC and UTC, respectively), on the four dates (2021/2022) at different soil depths.

Date	OTC			UTC		
Depth	High	Low	Signif.	High	Low	Signif.
17 September 2021:						
0–10 cm	3147 ± 1007	2773 ± 1045	*	3226 ± 1302	2497 ± 1247	**
10–20 cm	4207 ± 1001	3956 ± 1205	*	3491 ± 964	4199 ± 1153	**
20–30 cm	3231 ± 1226	4773 ± 1306	**	2068 ± 727	3369 ± 1125	**
0–30 cm	3602 ± 809	3246 ± 1105	*	3142 ± 1203	3033 ± 1102	ns
3 November 2021:						
0–10 cm	2413 ± 801	2599 ± 710	ns	2611 ± 734	2065 ± 518	**
10–20 cm	3092 ± 1332	2242 ± 543	**	2248 ± 268	2377 ± 630	ns
20–30 cm	2929 ± 1226	1974 ± 462	**	1878 ± 617	2164 ± 649	*
0–30 cm	2811 ± 1007	2281 ± 412	**	2253 ± 367	2202 ± 403	ns
21 December 2021:						
0–10 cm	2681 ± 581	2321 ± 387	*	2090 ± 862	1871 ± 459	*
10–20 cm	3215 ± 878	2366 ± 420	**	2419 ± 731	2623 ± 538	ns
20–30 cm	2634 ± 1570	1925 ± 572	**	2596 ± 1252	2586 ± 1012	ns
0–30 cm	2774 ± 619	2214 ± 316	**	2338 ± 857	2401 ± 438	ns
15 March 2022:						
0–10 cm	3226 ± 927	2320 ± 607	**	2251 ± 809	2439 ± 720	ns
10–20 cm	3658 ± 1157	2654 ± 613	**	3064 ± 1229	3101 ± 950	ns
20–30 cm	3016 ± 1172	2518 ± 773	**	3049 ± 1524	3316 ± 1023	*
0–30 cm	3240 ± 799	2506 ± 387	**	2706 ± 1010	2941 ± 628	ns

Signif.—Significance; **—Statistically significant at the 99% confidence level (*p* < 0.01); *—Statistically significant at the 95% confidence level (*p* < 0.05); ns—Not significant.

## Data Availability

Not applicable.

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
