# Peer review of "Sensing and Mapping the Effects of Cow Trampling on the Soil Compaction of the Montado Mediterranean Ecosystem"

_sensors, 2023, doi:10.3390/s23020888_

Round 1
Reviewer 1 Report
Review report: sensors-2116791
Manuscript entitled “A Technological Approach to Sensing and Mapping the Effects of Tree Roots and Cow Trampling on Soil Compaction of Montado Mediterranean Ecosystem” submitted Serrano et al. is dealing with quite interesting issue. The findings are good and could be interesting for researchers. However, following comments should be addressed before proceeding this manuscript for further. Authors are strongly advised to correct manuscript as per following suggestions for enhancing readability and reproducibility of results.
1. What is the novelty of the research? The necessity and innovation of the article should be presented in the last paragraph of introduction section.
2. Key words should be arranged in alphabetical order.
3. Abbreviations used throughput the manuscript needs to be defined the first time they are used.
4. In the method: We all known that soil properties are affected by geology and topographic features. Thus, I suggest you provide a figure of land-cover types.
5. In my opinion it would be better if you introduce the depth with cm instead of m.
6. The following references maybe helpful for this paper and recommended to be cited.
https://doi.org/10.1016/j.rse.2021.112321;https://doi.org/10.1007/s12665-021-09760-x; https://doi.org/10.1016/j.still.2021.105074
7. Authors are highly recommended to combining Result and discussion for enhancing readability and following it by the reader.
Author Response
REVIEWER#1
Comments and Suggestions for Authors
Review report: sensors-2116791
Manuscript entitled “A Technological Approach to Sensing and Mapping the Effects of Tree Roots and Cow Trampling on Soil Compaction of Montado Mediterranean Ecosystem” submitted Serrano et al. is dealing with quite interesting issue. The findings are good and could be interesting for researchers. However, following comments should be addressed before proceeding this manuscript for further. Authors are strongly advised to correct manuscript as per following suggestions for enhancing readability and reproducibility of results.
R- The authors are grateful for all of the reviewers' suggestions and comments, which contributed decisively to the improvement of the article.
- 1. What is the novelty of the research? The necessity and innovation of the article should be presented in the last paragraph of introduction section.
R- The last paragraph of “Introduction “ section (“This study aims:…”) already includes the reference to the interest of this study (“…(iii) to demonstrate the interest of combine various technological tools for sensing and mapping…”).
This paragraph was rewritten accordingly.
- Key words should be arranged in alphabetical order.
R- The reviewer suggestion was incorporated. Thank you.
- Abbreviations used throughput the manuscript needs to be defined the first time they are used.
R- The reviewer is right. The manuscript was reviewed accordingly. Thank you.
- In the method: We all known that soil properties are affected by geology and topographic features. Thus, I suggest you provide a figure of land-cover types.
R- We do not sure if we correctly understand the suggestion: it refers to general land-cover types? We feel that a generalist approach (to land-cover types) may distract readers from this case study.
- In my opinion it would be better if you introduce the depth with cm instead of m.
R- We used "m" because it is SI length base unit. We ask the reviewer to understand that to change now to "cm" would require redoing almost all the figures.
- The following references maybe helpful for this paper and recommended to be cited.
https://doi.org/10.1016/j.rse.2021.112321; https://doi.org/10.1007/s12665-021-09760-x; https://doi.org/10.1016/j.still.2021.105074
R- The reviewer's suggestion was accepted. A new citation (Xu et al., 2021) was incorporated in the manuscript.
- Authors are highly recommended to combining Result and discussion for enhancing readability and following it by the reader.
R- The reviewer is right. The manuscript was reviewed accordingly. Thank you.

Reviewer 2 Report
Dear authors,
Read my suggestions, which are given in the attached file.

Author Response
REVIEWER#2
Comments and Suggestions for Authors
Read my suggestions, which are given in the attached file.
R- The authors are grateful for all of the reviewers' suggestions and comments, which contributed decisively to the improvement of the article.
1- (Title) The title must be changed and brought into line with the established objectives of the research. It is too long and inappropriate. I didn't find any observations of tree roots or results that were related to tree roots. Furthermore, what technical approach did you take in this paper that was not previously known? You just combine several sensing technologies during the trial.
R- The reviewer's suggestion was accepted: the title was simplified accordingly to “Sensing and Mapping the Effects of Cow Trampling on Soil Compaction of Montado Mediterranean Ecosystem”.
2- (Objectives) It is very interesting how you will evaluate tree root impact on the compaction.
R- As in the title, the reference to "tree root impact" was removed from the objective (ii) (in the Abstract and Introduction).
3-(Figure 3) Do the red squares on the image lay exactly over the Sentinel pixels or not? If not, use points instead of squares for the Sentinel pixel center.
R- Yes, as mentioned in the manuscript (2.1. Section: “…Sentinel-2 pixels “10 m × 10 m” were georeferenced for sampling in areas without trees…”), the sampling was based on Sentinel pixels.
4- You should, in my opinion, take a lot more CI measurements on a 10 x 10 m area not just five.
R- We understand the reviewer's observation: accuracy increases with more measurements (repetitions), however, we sought a compromise that would allow reliable results to be obtained without requiring excessive effort in the field determination. We took as our guide the suggestions of Pias et al. (2018- Reference 19): “A reduction in the number of subsamples promoted an increase in the variability of the data. Generally, the results from this study suggest the use of at least four subsamples per sampling point to achieve maps with a coefficient of relative deviation less than 10% and significant correlation with the reference maps.”
19- Pias, O.H.C.; Cherubin, M.R.; Basso, C.J.; Santi, A.L.; Molin, J.P.; Bayer, C. Soil penetration resistance mapping quality: effect of the number of subsamples. Acta Scientiarum 2018, 40, e34989.
5- Where are the maps?
R- The map is in the “Results” section (Figure 15).
6- (Table 1) Why was NDVI omitted here?
R- The soil parameters (namely the CI) summarized in Table 1 were measured only at certain moments (in 2018 and 2019), while the NDVI time series covered one year (between July 2021 and June 2022), with a temporal resolution of 5 days, so if they were considered in this table, it would become very extensive. The practical interest of NDVI is in its temporal evolution, represented in Figure 17.
7- (Figure 6) Which geostatistical or deterministic approach did you use for mapping? I would like to see a description in the materials and methods section. Also, I would like to see the results of spatial modelling
R- The reviewer's suggestion was accepted: a description of soil maps elaboration was included in the Material and Methods (2.4. section).
8- (Figure 7) Why didn't you include the NDVI map? If you map soil properties, you should include NDVI.
R- As mentioned above, soil parameters were measured only at certain moments, while the NDVI time series covered one year (between July 2021 and June 2022), with a temporal resolution of 5 days, so if they were fully considered in maps, they would make the article too long and complex.
9- (Table 2) You missed NDVI?
R- Please see the previous response.
10- (Figure 8) If you correlate clay with EC, you don't need to do that for sand and EC because these two variables have direct-inverse proportions over space and provide direct-inverse results with no additional informative value.
R- The reviewer is right when referring to the known inverse relationship of clay and sand with EC. However, using only clay or clay and sand implies the same graphical space in Figure 8. Given that there are not many studies in our region between soil properties and EC, it seems to us that this information may be useful for future work. We thank you for your comment. Please allow us to put it to the editor's consideration.
11- (Figure 9) This is a very strange chart. You have a positive relationship between SMC and CI, which is quite unusual. The same holds true for the EC-SMC relationship. It is normal for soils that contain a higher level of moisture to have higher electrical conductivity, and vice versa.
R- We do not understand the reviewer's point: contrary to what the reviewer states, the graph of Figure 9 does not show a positive relationship between SMC and CI, but a negative relationship, as one would expect!
12- (Pg. 13) Do you mean tree location? How did you assess the impact of tree roots on soil properties?
R- The references to the impact of tree roots on soil properties were removed from the manuscript.
13- (Figure 14) This result is contradictory to the one you presented in figure 9. How is it possible—the same soil, the same measurement method, opposite results?
R- As mentioned above in relation to Figure 9, the results (Figure 9 and Figure 14) are concordant and not contradictory: in both figures the relationship between EC and SMC is negative, as expected.
14- (Figure 15) Why did you choose to use outlier analysis? Why didn't you just use GPS data to interpolate? Why isn't the raster resolution the same as it was on previous maps, or is it?
R- This type of analysis (“Outlier Analysis”) allows to detect “High” and “Low” clusters in terms of animal density, but at the same time allows to detect “Outliers” inside of each cluster or saying in another way, high concentration points of animals inside of low animal density clusters and low concentration points of animals inside of high animal density clusters. This type of probabilistic analysis will not be possible to develop in a normal interpolation algorithm. The raster spatial resolution was, in this case, of 5 m. As suggested by the reviewer, this information was included in the manuscript (2.6 section).
15- Tables 4 and 5 are tricky to understand, especially due to the separate data representation over the soil layers.
R- R- We understand the reviewer's point because a large amount of field data is presented. However, the use of colours both to identify zones of "low" and "high" animal trampling and to identify zones of higher (darker) and lower (lighter) CI was the way we considered to make these tables easier to read and interpret.
16- (4.2 Section) Here, you mention the impact of soil compaction on root development, and in the title, you mention the effect of tree roots on soil compaction. This is in collision, I think.
R- The reviewer is right. As suggested, the reference to root development was removed from the manuscript. The text was rewritten accordingly.
17- (4.2 Section) This is not a discussion of the results.
R- The reviewer is right. As suggested, this sentence was removed.
18- (4.2 Section) This is already mentioned.
R- The reviewer is right. As suggested, this sentence was removed.
19- (4.2 Section) This is not main reason, definitively. Increased soil moisture reduces the cohesive forces between clay particles; consequently, less energy is required for tillage as there is less resistance (https://doi.org/10.1007/s11119-021-09805-y).
R- The reviewer is right. The suggestion (new reference: Kostic et al., 2021) was incorporated in the manuscript. Thank you.
20- (4.3 Section) Cone Index Spatial Variability and Livestock Trampling .... should be considered together, not separately. Both are related to soil compaction.
R- We respect the reviewer's opinion and agree that both, Cone Index Spatial Variability and Livestock Trampling, are related to soil compaction, however if the discussion of the results were done together it would be much more complex for the reader of the journal.
21- (4.3 Section) This is not a discussion of the results. You have many excessive sentences that just extend the manuscript's text without adding to its quality.
R- The reviewer is right. As suggested the text was rewritten accordingly.
22- (4.4 Section) This is not a discussion of the results.
R- The reviewer is right. The text was rewritten accordingly.
23- (4. 4 Section) Which facts support your claim that grazing management undoubtedly contributed to vegetation if the presence of various plant species on tested fields is highly spatially variable? Your statements have weak data support.
R- We are not sure if we correctly understand the question. The justification for the behaviour of NDVI (Figure 17), in particular for the significant differences in NDVI between the low and high livestock trampling sampling points due to the effect of "(i) differential grazing management" is centred on the “differential grazing management” of fields A and B. It is explained in the manuscript that Field A (where “high livestock trampling” areas predominate) was, in the period referred to (March), left as rest (not grazed), contrary to Field B (where “low livestock trampling” areas predominate), an aspect with a natural impact on the vegetative vigour of the pasture, independently of the high spatial variability of plant species.
24- (4.5 Section) This is not a discussion of the results.
R- The reviewer is right. As suggested this sentence was removed from the manuscript.
